A synthetic defective interfering SARS-CoV-2

Yao Shun 1
Narayanan Anoop 2
Majowicz Sydney A. 2
Jose Joyce jxj321@psu.edu 2 3
Archetti Marco mua972@psu.edu 1 3
1 Department of Biology, Pennsylvania State University , University Park , United States of America
2 Department of Biochemistry & Molecular Biology, Pennsylvania State University , University Park , United States of America
3 The Huck Institutes for the Life Sciences, Pennsylvania State University , University Park , United States of America
Gomez Shawn
Electronic publication date: 2021 Jul 1
Publication date: 2021
Volume: 9
Electronic Location ID: e11686
Received 2021 Mar 23; Accepted 2021 Jun 7
Copyright: ©2021 Yao et al.
Copyright year: 2021
Copyright holder: Yao et al.
License: This is an open access article distributed under the terms of the Creative Commons Attribution License, which permits unrestricted use, distribution, reproduction and adaptation in any medium and for any purpose provided that it is properly attributed. For attribution, the original author(s), title, publication source (PeerJ) and either DOI or URL of the article must be cited.
License URL: https://creativecommons.org/licenses/by/4.0/

Keywords: Covid-19, SARS-CoV-2, Defective Interfering Particle, Synthetic Biology, Coronavirus

Funding: The Pennsylvania State University A Huck Institutes for the Life Sciences COVID-19 seed grant This work was supported by the Pennsylvania State University and a Huck Institutes for the Life Sciences COVID-19 seed grant. The funders had no role in study design, data collection and analysis, decision to publish, or preparation of the manuscript.

==============================
Viruses thrive by exploiting the cells they infect, but in order to replicate and infect other cells they must produce viral proteins. As a result, viruses are also susceptible to exploitation by defective versions of themselves that do not produce such proteins. A defective viral genome with deletions in protein-coding genes could still replicate in cells coinfected with full-length viruses. Such a defective genome could even replicate faster due to its shorter size, interfering with the replication of the virus. We have created a synthetic defective interfering version of SARS-CoV-2, the virus causing the Covid-19 pandemic, assembling parts of the viral genome that do not code for any functional protein but enable the genome to be replicated and packaged. This synthetic defective genome replicates three times faster than SARS-CoV-2 in coinfected cells, and interferes with it, reducing the viral load of infected cells by half in 24 hours. The synthetic genome is transmitted as efficiently as the full-length genome, suggesting the location of the putative packaging signal of SARS-CoV-2. A version of such a synthetic construct could be used as a self-promoting antiviral therapy: by enabling replication of the synthetic genome, the virus would promote its own demise.

Introduction

Versions of a viral genome with large deletions arise frequently from most RNA viruses when passaged in vitro (Gard et al., 1952; Huang & Baltimore, 1970; Brian & Spaan, 1997; Vignuzzi & López, 2019). Defective genomes lacking essential coding sequences can still replicate and be packaged into virions in the presence of functional full-length viruses. The full viral genome produces the essential proteins for replication and packaging, which can be exploited by defective genomes that retain the ability to bind to these proteins. These defective genomes can be considered parasites of the full-length virus, as they compete for replication and packaging and, given their shorter length, can replicate faster than (and interfere with) their parental full-length viral genome in coinfected cells.

Such defective interfering (DI) genomes have been described–and indeed appear to be common–in coronaviruses (Kim, Jeong & Makino, 1993; Kim, Lai & Makino, 1993; Méndez et al., 1996; Brian & Spaan, 1997; Izeta et al., 1999; Graham et al., 2006; Fehr & Perlman, 2015; Vignuzzi & López, 2019), where they have been used to locate the functional elements of their genomes. In SARS-CoV-2, the virus responsible for the current Covid-19 pandemic (Wu et al., 2020; Zou et al., 2020), long deletions have been reported (Kim et al., 2020), and DI genomes have been shown to arise by recombination driven by sequence microhomology (Gribble et al., 2021).

We made short synthetic DI RNAs from parts of the SARS-CoV-2 genome to test whether these DI genomes could replicate in coinfected cells and be packaged into virions. If our DI genomes replicate faster than the wild type (WT) virus genome, the DIs could impair the intracellular growth of the virus, and if the DI genomes get packaged efficiently into virions, this interference could continue over time.

The design of our DI genomes was based on observations from natural defective interfering coronaviruses (TGEV: Méndez et al., 1996; MHV: Makino, Fujioka & Fujiwara, 1985; Makino et al., 1988; Makino, Yokomori & Lai, 1990; Van der Most, Bredenbeek & Spaan, 1991; Kim, Jeong & Makino, 1993; Kim & Makino, 1995; Masterset al., 1994; Goebel et al., 2007; BCoV: Chang et al., 1994; Chang & Brian, 1996; Williams, Chang & Brian, 1999; Raman et al., 2003; Brown et al., 2007; 229E: Thiel, Siddell & Herold, 1998; IBV: Penzes et al., 1994, Dalton et al. 1998; reviewed by Yang & Leibowitz, 2015) suggesting that the 3′ and 5′ untranslated regions (UTRs) are essential for replication and that the putative packaging signal resides inside the nsp15 sequence (TGEV: Escors et al., 2003; Morales et al., 2013; Hsieh et al., 2005; Hsin et al., 2018; MHV: Fosmire, Hwang & Makino, 1992; Kuo & Masters, 2013; Woo et al., 2019; BCoV: Cologna & Hogue, 2000)—a conclusion that is, however, disputed for A betacoronaviruses, which lack the RNA structure responsible for packaging (Masters, 2019). DI genomes that occur naturally in SARS-CoV-2 often retain the 5′ UTR and 3′ UTR; 80% of these DIs have single deletions; the most abundant DI genomes with double deletions have a very short terminal deletion and a long central one (Gribble et al., 2021).

Our main synthetic construct is made from three portions (Fig. 1): the 5′ UTR and the adjacent 5′ part of nsp1 in ORF1a, a part of nsp15 that includes the putative packaging signal, and the sequence spanning the 3′ part of the N sequence, ORF10 and the 3′ UTR. We chose the N fragment to include two of the most conserved regions of the virus genome (Rangan et al., 2020) (28990–29054 and 28554–28569). Because there is evidence that a long ORF enables DIs in certain coronaviruses (notably MHV (DeGroot, Most & Spaan, 1992), which is closely related to SARS-CoV-2) to replicate more efficiently (even if coding for a chimeric non-functional protein: Van der Most et al., 1995), we assembled the three fragments in frame, to retain a 2247 nt ORF starting at the 5′ end of nsp1 (Fig. 1); and because there is evidence (Joo & Makino, 1995; Van Marle et al., 1995; Méndez et al., 1996) that multiple transcriptional regulatory sequences (TRS) reduce replication efficiency, we chose the 3′ portion to start from within the N sequence, to exclude its TRS. Analysis of the predicted secondary structure of this synthetic RNA showed that the three portions fold essentially (except at the junctions) like the corresponding sequences in the full genome.

Figure 1 Synthetic defective interfering viruses.

(A) Three portions of the wild type (WT) SARS-CoV-2 genome were used to create a synthetic defective interfering genome (DI1) and a shorter version (DI0) comprising only parts of the two terminal portions. Numbers delimiting the portions refer to positions in the SARS-CoV-2 genome. The first position is mutated (A →C) in both DI1 and DI0. Open rectangles show the position of the probes and primers used. (B) To produce synthetic DI particles, DNA constructs corresponding to the RNA sequence of DI1 or DI0 were transcribed into RNA in vitro using T7 RNA polymerase and transfected into Vero-E6 cells that were then infected with SARS-CoV-2. The supernatant from these cell cultures was used to infect new cells.

The length of our main synthetic DI genome (DI1) is 2882 nt, 9.6% of the full-length genome (29903 nt). We also synthesised a shorter (800 nt) defective genome (DI0) without the second portion (the putative packaging signal) and with shorter terminal portions (Fig. 1) as control and to test the effects of the intersecting portions on replication and packaging. The DI1 and DI0 genomes, synthesised as DNA and cloned into plasmids, were transcribed in vitro to produce genomic RNAs, which were then electroporated in Vero-E6 cells, which were subsequently infected with SARS-CoV-2.

Materials & Methods

Sequences and cloning

The DNA sequence of the DI1 genome (GenBank accession number: MW250351) was designed to correspond to the following three joint portions of the SARS-CoV-2 complete genome (the NCBI Reference Sequence for SARS-CoV-2; GenBank accession number: NC_045512.2), in the following order: 1 to 789; 19674 to 20340; and 28478 to 29903. The DI0 genome (GenBank accession number: MW250350) was designed to correspond to the following two joint fragments of SARS-CoV-2 in the following order: 1 to 473; 29576 to 29903. In both cases, the first nucleotide of the first fragment was changed from A to C to improve in vitro transcription efficiency (Milligan et al., 1987; Martin & Coleman, 1987). The synthetic sequence was analysed using the Vienna RNA package (Lorenz et al., 2011) to confirm the absence of potential aberrations in the RNA secondary structure. The DI1 and DI0 genome DNA were assembled from synthetic oligonucleotides and inserted into a pMA-RQ plasmid by Invitrogen (Thermo Fisher Scientific). The minimal T7 promoter TAATACGACTCACTATAGG was synthesised immediately upstream of the 5′ end of the synthetic virus sequence. A short sequence (CCATGG) containing the NcoI restriction site was synthesised immediately upstream of the 5′ end of the T7 promoter, and a short sequence (CCGGT) containing the AgeI restriction site was synthesised immediately downstream of the 3′ end of the third fragment. The plasmid DNA was purified from transformed bacteria and the final construct was verified by sequencing.

In vitro transcription

The plasmid containing the synthetic DI1 or DI0 genome DNA was linearized using NcoI and AgeI and resuspended in H2O. 1 µg was then used as a template to produce capped RNA via T7 RNA polymerase, using a single reaction setup of the mMESSAGE mMACHINE® Kit (Applied Biosystems), which contains: 2 µL enzyme mix (buffered 50% glycerol containing RNA polymerase, RNase inhibitor, and other components); 2 µL reaction buffer (salts, buffer, dithiothreitol, and other ingredients); 10 µL of a neutralized buffered solution containing: 15 mM ATP, 15 mM CTP, 15 mM UTP, 3 mM GTP and 12 mM cap analog [m7G(5′)ppp(5′)G]; 4 µL nuclease-free H2O; incubated for 2 h at 37 °C. RNA was isolated using TRIzol reagent (Invitrogen) extraction and isopropanol precipitation.

Cells and transfection

Vero-E6 cells (ATCC: CRL-1586) cultured in DMEM medium (Hyclone, #SH30022.FS) supplemented with 10% fetal bovine serum (Corning, #35-011-CV), 100 units ml−1 penicillin and 100 µg ml−1 streptomycin (Gibco, #15140122) maintained at 37 °C and in a 5% CO2 atmosphere were grown to 80% confluence. The cells were electroporated with the RNA produced by in vitro transcription (DI1: 532 ng; DI0: 476 ng; per 200,000 cells; equivalent to 1.7 × 106 and 5.6 × 106 RNA molecules per cell, respectively), in 100 µl Nucleocuvette Vessels using the SF Cell solution and program DN-100 on a 4D Nucleofector X unit (Lonza). The efficiency of transfection was approximately 90% for both the DI1 and DI0 RNAs. Cells used for the control experiments were electroporated in the same way but without RNA.

Virus culture

SARS-CoV-2 isolate USA-WA1/2020 was obtained from BEI resources (#NR-52281) and propagated in Vero-E6 cells. Virus stocks were prepared, and the titer determined by plaque assays by serially diluting virus stock on Vero-E6 monolayers in the wells of a 24-well plate (Greiner bio-one, #662160). The plates were incubated at room temperature in a laminar flow hood with hand rocking every ten minutes. After one hour, an overlay medium containing 1XMEM, 1% Cellulose (Millipore Sigma, #435244), 2% FBS and 10 mM Hepes 7.5, was added and the plates were incubated for a further 48 h at 37 °C. The plaques were visualized by standard crystal violet staining. All work with SARS-CoV-2 was conducted in Biosafety Level-3 conditions at the Eva J Pell Laboratory of Advanced Biological Research, The Pennsylvania State University, following the guidelines approved by the Institutional Biosafety Committee (IBC# 48724).

Coinfection and RNA extraction

200,000 transfected cells were seeded in each well of a 24-well plate (each well in triplicate; except the 24 h time-point of the initial coinfection, with 12 replicates) and incubated for 1 h before being inoculated with SARS-CoV-2 at MOI = 10. The medium containing the infectious SARS-CoV-2 viruses was removed after 1 h and replaced with fresh medium. Cells were allowed to grow for 4, 8, 12 or 24 h before RNA was extracted. The supernatant of cultures grown for 24 h was used to infect new cells in 24-well plates for one hour, then media were replaced with fresh media and RNA was extracted from the cells after another 24 h. This step was repeated four times to obtain RNA from four consecutive passages. RNA was extracted using Quick RNA miniprep kit (Zymo, #R1055) or TRIzol reagent (Invitrogen, #15596026) followed by isopropanol precipitation.

RNA analysis

Equal amounts of total RNA were reverse transcribed into first-strand cDNA using Revert Aid First Strand cDNA Synthesis Kit (Fermentas). 2 µl diluted cDNA (3pg-100ng depending on the experiment), 2 µl of 5 µM primer mix (forward plus reverse), 1 µl of 2 µM of probe and 5 µl master mix (2 ×) was used for qRT-PCR using TaqMan assay on a StepOnePlus instrument (Applied Biosystems) starting with polymerase activation at 95 °C for 3 min, followed by 40 cycles of denaturation (95 °C, 15 s) and annealing/extension (60 °C, 1 min). The amount of WT and synthetic DI1 or DI0 genomes were quantified (using StepOnePlus Software 2.3) by the comparative CT method (Livak & Schmittgen, 2001). All results were normalised with reference to the actin beta (ACTB) gene of Chlorocebus sabaeus; each sample was repeated three times and the average value was used; all absolute values reported are 2−ΔCT values. Primers and probes for the DI1 and DI0 genomes were designed to amplify one of the junctions between portions of the WT genome; the probe was designed to span a junction that is not present in the WT genome (Fig. 1) and is unlikely to be found in naturally occurring DIs. Our DI primer-probe sets gave consistently negative results in qPCR tests of virus-only control plates. For the virus we used a modified version of the CCDC primer-probe set on ORF1. A BLAST search revealed no off-target sequences in the SARS-CoV-2 or in the Chlorocebus sabaeus genome. Primers and probes were labelled using the FAM dye, an IBFQ quencher and an additional internal (ZEN) quencher and were synthesised by Integrated DNA Technologies. The sequences are the following.

DI1 forward: 5′-AGCTTGGCACTGATCCTTATG-3′

DI1 reverse: 5′-ACATCAACACCATCAACTTTTGTG-3′

DI1 probe: 5′-FAM/TTACCCGTGAACTCATGCGACAGG/IBFQ-3′

DI0 forward: 5′- ATCAGAGGCACGTCAACATC -3′

DI0 reverse: 5′- TTCATTCTGCACAAGAGTAGACT -3′

DI0 probe: 5′-FAM/ AGCCCTATGTGTCGCTTTTCCGT /IBFQ-3′

SARS-CoV-2 forward: 5′- CCCTGTGGGTTTTACACTTAA -3′

SARS-CoV-2 reverse: 5′- ACGATTGTGCATCAGCTGA -3′

SARS-CoV-2 probe: 5′-FAM/CCGTCTGCGGTATGTGGAAAGGTTATG /IBFQ-3′

ACTB forward: 5′-AGGATTCATATGTGGGCGATG-3′

ACTB reverse: 5′-AGCTCATTGTAGAAGGTGTGG-3′

ACTB probe: 5′-FAM/AGCACGGCATCGTCACCAACT/IBFQ-3′

We were able to quantify the relative amounts of DI and WT genomes by qRT-PCR, hence we report their relative values across time points or treatments; but since we use different primer-probe sets for the DI and WT genomes, we cannot compare the absolute values of the DI and WT genomes, hence we cannot measure the DI/WT ratio. For each genome g, however, if we define the 2−ΔCT value at time point i as 2−ΔCT(g,i), we can measure the ratio of 2−ΔCT(g,i) values at two different time points i =t 1,t2 as Rg(t1,t2) =2−ΔCT(g,t1)/2−ΔCT(g,t2). The ratio ρ(t 1,t 2) = RDI(t1,t 2)/ RWT(t1,t2) reveals the rate of increase of the DI genome across time points (t1,t2), relative to the increase of the WT genome across the same time points.

Results

Because of the fast degradation of the DI RNA inside cells (in the absence of the virus, 1 to 4% of the initial synthetic RNA can be detected by qRT-PCR 4 h post transfection) and the lag between infection and viral protein production, it is not possible to quantify the replication rate of the DI1 and DI0 genomes, or even prove their replication, immediately after RNA transfection. It is possible, however, to quantify interference of the DI RNA with the WT virus in coinfections: within 24 h of transfection, the DI1 genome reduced the amount of SARS-CoV-2 by approximately half compared to the amount of virus in control experiments (Welch’s unequal variances t-test: t15.3 = 3.18, p = 0.006). The DI0 genome, instead, had no significant interference effect (Fig. 2A) (Welch’s unequal variances t-test: t2.1 =−0.4, p =0.72), which, in addition to serving as a control for the effect of DI particles, suggests that the parts of the DI1 genome missing in the DI0 genome are essential for replication.

Figure 2 DI1 reduces the amount of SARS-CoV-2 by half; it replicates 3 times faster; and it is transmitted with the same efficiency.

(A) Growth rates (absolute amount relative to the amount at 4 h) of WT in controls (gray) and in coinfections with DI1 (blue) or DI0 (green); growth relative to controls at the same time point; and detail at 24 h. (B) Transmission efficiency of WT (blue) and DI1 (yellow) in coinfections: the amount, measured by qRT-PCR, immediately before passaging divided by the average amount measured almost immediately (4 h) after passaging (using the supernatant to infect new cells 24 h after initial infection). DI0 was detected inside the cells but not in the supernatant. (C) Growth rates (absolute amount relative to the amount at 4 h) of WT in controls (gray) and in coinfections (blue); growth relative to controls at the same time point; and detail at 24 h. Growth rates (absolute amount relative to the amount at 4 h) of WT (blue) and DI1 (yellow) in coinfections; growth relative to that of WT in coinfections at the same time point; and detail at 24 h.

24 h post transfection the supernatants were collected and used to infect new cell monolayers. In these cells we detected (by qRT-PCR) the DI1 and WT genomes, from 4 to 24 h after the transfer. The DI0 genome was not detected, suggesting that the parts of the DI1 genome missing in the DI0 genome have a positive effect on packaging. The transmission rate of the DI1 genome did not differ from that of the WT genome (Fig. 2B) (Student’s t-test: t22 =0.49, p = 0.62), suggesting that the synthetic genome gets packaged into viral particles with essentially the same efficiency as the full-length virus, and that these viral particles are as infectious. In these cells coinfected by DI1 RNA and SARS-CoV-2, the WT genome again declined by approximately half in 24 h (Fig. 2C) (Student’s t-test: t4 =2.95, p = 0.042). The replication rate of the DI1 genome could now be quantified, revealing that it increases 3.3 times as fast as the WT virus (Fig. 2C) (Student’s t-test: t4 =−2.74, p = 0.052).

Since the packaging efficiencies of the DI1 genome and of the WT genome are not significantly different, we can rule out the possibility that this observed relative increase in intracellular DI1 RNA is due to its lower rate of packaging. And since the supernatant from the previous passage was removed 1 h after infection, the increase in DI1 RNA we observed is likely due to replication. Since the intracellular DI1 genomes in this second passage derive entirely from infectious viral particles produced in the first passage, we can conclude (in addition to our previous observation that the control DI0 RNA does not interfere with replication) that the reduction in the amount of WT observed in coinfections (here and in the first passage) is due to interference brought about by the faster replication of the DI1 genome, and not to any collateral effect of the initial transfection process.

We repeated the procedure by transferring the supernatant to new cells, coinfected with the WT virus, after 24 h for four times. The DI1 genome was detected across all four passages and while we were unable to measure the absolute WT/DI1 RNA ratio (because the amount of DI1 RNA was below the level detectable by digital PCR), the ρ(t 1,t 2) value increased approximately 3 times at every passage, consistent with the relative replication advantage and equal transmission efficiency we measured. Our results, therefore, suggest that even a small amount of DI1 RNA (small enough to be below the detection limit of digital PCR, but high enough to be detectable by qRT-PCR) can interfere with the WT virus.

Discussion

DI particles have long been known to virologists (Gard et al., 1952; Huang & Baltimore, 1970) and their use in unravelling the location of functional elements of a genome is well known. Our synthetic DIs suggest that a disputed (Masters, 2019) putative packaging sequence of SARS-CoV-2 could indeed enable packaging of our synthetic defective genome –and therefore presumably acts as a packaging signal for the WT genome. However, because the difference between our DI1 and DI0 synthetic constructs is not limited to the portion with the putative packaging signal (part of nsp15), we cannot rule out that packaging signals reside in the other parts of the DI1 genome that DI0 lacks, most notably a conserved region (28554–28569) with a SL5 motifs in the N partial sequence included in the DI1 genome but not in the DI0 genome. It is also possible that DI0 can be packaged but because it does not replicate efficiently, it is rapidly degraded after transfection and the amount of packaging does not meet the threshold for detection.

The interference with the WT virus is the most remarkable effect of our DI1 construct. As we have shown, while DI0 does not interfere significantly with WT, DI1 induces a reduction of about 50% in the amount of WT virus in coinfections compared to infections with WT alone, and this is likely due to the faster replication of the DI1 genome. DI particles are often described as by-products of inaccurate replication or as having a regulatory function for a viral quasi-species. However, DIs can also be seen as defectors in the sense of evolutionary game theory (Szathmáry, 1994; Turner & Chao, 1999, Brown, 2001): ultra-selfish replicators, able to freeride as parasites of the full-length genome. As such, DI particles need not serve any purpose for the WT virus.

Indeed, DIs could be used as antivirals: by virtue of their faster replication in cells coinfected with the WT virus, DI genomes can interfere with the virus. Potentially, as the DI genomes increase in frequency among the virus particles pool, the process becomes more and more effective, until the decline in the amount of WT virus leads to the demise of both virus and DI. A similar therapeutic approach has been proposed for bacteria (Brown et al., 2009) and cancer (Archetti, 2013). The potential of DIs as antivirals has been suggested before (Marriott & Dimmock, 2010; Dimmock & Easton, 2014; Vignuzzi & López, 2019), and a synthetic DI particle could perhaps be immune to the evolution of resistance (although coevolution of viruses and DIs has been shown in Rhabdoviridae (Horodyski, Nichol & Spindler, 1983). Unlike, for example, HIV and influenza, which are perhaps not ideal candidates because of their short genome, multiple genomic fragments and complex replication process, coronaviruses may be more amenable to DI therapy because of their long single fragment genome and relatively simple life cycles. While the immediate 50% reduction in virus load we observed is arguably not enough for therapeutic purposes, efficacy would compound over time (as the DIs increase in frequency) and a higher initial efficacy could be obtained using a delivery vector and an improved version of the DI genome.

Conclusions

We have established a proof of principle that a synthetic defective interfering SARS-CoV-2 can replicate in cells infected with the virus and interfere with its replication. Further experiments are needed to verify the potential of SARS-CoV-2 DIs as antivirals. Our experiments should be repeated in human lung cell lines, against other variants of SARS-CoV-2 and by transfecting DI RNA after infection, a more realistic simulation of therapy, which will, however, ultimately require in vivo experiments. An efficient delivery method should be devised to increase the initial amount of DI RNA and to deliver it in vivo. It would also be interesting to measure how the fraction of DI and WT genomes changes over time to test whether the DI genome drives the WT genome to extinction, or they coexist at a mixed equilibrium. Finally, it would be useful to analyse the long-term evolution of coinfections to test how SARS-CoV-2 and its DIs coevolve and whether resistant mutants can arise.

Supplemental Information

Supplemental Information 1 Growth rates in coinfections and control experiments reported in Fig. 2

Growth rates (absolute amount relative to the amount at 4 h) of WT in controls and in coinfections with DI or DI0; growth relative to controls at the same time point. Growth rates (absolute amount relative to the amount at 4 h) of WT in controls and in coinfections; growth relative to controls at the same time point; and detail at 24 h. Growth rates (absolute amount relative to the amount at 4 h) of WT and DI in coinfections; growth relative to that of WT in coinfections at the same time point; and detail at 24 h.

Click here for additional data file.

Additional Information and Declarations

Competing Interests

Author Contributions

Ethics

Patent Disclosures

DNA Deposition

Data Availability

Marco Archetti is the inventor of a related pending patent application owned by the Pennsylvania State University.

Shun Yao performed the experiments, analyzed the data, authored or reviewed drafts of the paper, designed and performed cloning, in vitro transcription and transfection; designed and performed RT-qPCR analysis; analysed data, and approved the final draft.

Anoop Narayanan performed the experiments, authored or reviewed drafts of the paper, planned and performed virus culture and infection and RNA collection, and approved the final draft.

Sydney A. Majowicz performed the experiments, authored or reviewed drafts of the paper, performed virus culture and infection and RNA collection, and approved the final draft.

Joyce Jose analyzed the data, authored or reviewed drafts of the paper, planned and coordinated virus culture and infection and RNA collection, and approved the final draft.

Marco Archetti conceived and designed the experiments, performed the experiments, analyzed the data, prepared figures and/or tables, authored or reviewed drafts of the paper, conceived and coordinated the project; designed sequences, cloning and in vitro transcription; analysed data, and approved the final draft.

The following information was supplied relating to ethical approvals (i.e., approving body and any reference numbers):

All work with SARS-CoV-2 was conducted in Biosafety Level-3 conditions at the Eva J Pell Laboratory of Advanced Biological Research, The Pennsylvania State University, following the guidelines approved by the Institutional Biosafety Committee of The Pennsylvania State University (IBC# 48724).

The following patent dependencies were disclosed by the authors:

Penn State University submitted a preliminary patent application that mentions the DI construct we describe in the paper. The provisional patent numbers are US 63/060,327 and US 63/116,372.

The following information was supplied regarding the deposition of DNA sequences:

The defective interfering genome sequences are available at GenBank: MW250350 and MW250351.

The following information was supplied regarding data availability:

The raw data with the results described in Fig. 2 are available in the Supplemental File.

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
