# Peer review of "A synthetic defective interfering SARS-CoV-2"

_PeerJ, doi:10.7717/peerj.11686_

## Round 0.1 · original submission · Major Revisions

Thank you again for your submission. While the reviewers were generally positive, they did have a number of comments and questions that should be addressed as part of the revision. Please address these concerns as appropriate.

Reviewer 1 ·

Basic reporting

Overall, the manuscript is very well written. The introduction was clear and covered the most important literature.

Specific comments:
• For line 49, the authors should cite this paper (https://doi.org/10.1371/journal.ppat.1009226) as it has been recently shown that defective viral genomes can be found in SARS-CoV-2. It would be interesting to compare the DI made by the authors to the DIs found naturally in the paper above.
• For line 77-78: Just to clarify…were any aberrations found?
• The resolution of figures was not great.

Experimental design

The concept of using DIs as an antiviral is not novel by itself and has been applied to many viruses already but not too much work has been done yet on SARS-CoV-2. So this manuscript is a good addition to the antiviral DI literature. Overall, the data looks very convincing but there are lots of missing gaps that would make it more convincing that DIs could be used as an antiviral against SARS-CoV-2.

Specific comments:
• Have the authors tested the DI on another cell line? VERO-E6 are not an ideal model as not human, not lung and not having “normal” type I IFN signaling. It would be interesting to see if this DI would be even more interfering in a cell line that has functional type I IFN signaling. DIs have been shown that they can also be immunostimulatory.
• The DI was tested on one SARS-CoV-2 variant. As they are more and more variants have the authors tested if the DI would also be inhibitory against other variants? Is the effect specific to this variant?
• Virus have been shown to evolve to become resistant to DIs (https://doi.org/10.1016/0092-8674(83)90022-3). Is this something authors have observed throughout passages? If sequencing was done, did any new mutations appear?
• The antiviral effect of DI works when DIs are added since the beginning of the infection based on the authors experiments. This is of course not a very realistic scenario in patients. It would be interesting to see if the authors would add the DI at different time points if it affects the antiviral ability of the DI.
• DI have been shown to be antiviral short term but also long term to cause persistence. The authors state that virus is still present in their last passage. Have the authors done more passage and observed if the virus disappears completely at some point? Important to address this as line 255-257 DIs are only portrayed as antiviral.

Validity of the findings

It is encouraging to see that the authors were able to demonstrate that the DI as antiviral approach might also work for SARS-COV-2. Overall, current experiments are well designed but some clarifications are needed.

Specific comments:
• For passage 1, the authors could have generated purified DI particles (DIPs) stocks instead of doing transfection (for example: https://doi.org/10.1093/infdis/jiz564). Having DIs as purified DIPs form would make it easier (as combination of transfection and infection can be very challenging sometimes) also to answer more questions such as the time point comment in the previous section.
• An MOI of 10 was used for original infection. DIs generally are generated at high MOI. Do the authors have any concern that in addition to synthetic DIs you might also detect naturally occurring DIs?
• What did the author do to make sure the different primers have about the same efficiency? This is important as this could bias results.
• Figure 1:Based on methods all these experiments were done in triplicates, but most conditions have more of different amounts of dots. What do these represent? Biological repeats? Technical repeats? What type of statistic was use?
• Line 224-226: This sentence “We were unable to measure the absolute WT/DI ratio because the amount of DI was below the level detectable by digital PCR.” is unclear. How were authors able to make graphs if DI not detectable? Please clarify.

Reviewer 2 ·

Basic reporting

no comment

Experimental design

no comment

Validity of the findings

1. It is unclear from the results whether DI_0 genomes could be detected at any time after transfection. Is it possible that despite their relatively small size that they were unable to be transfected into cells?

2. Lines 145-148, 221-227. It was noted that the DI genome was detected across all four passages (Line 222), but the amount of DI was below the level detectable (Line 225); these statements appear to be contradictory.  Was there a monotonic increase in DI genomes or monotonic decrease of WT genomes across the four passages? It is unclear at what point (which passage) the DI and/or WT genomes fell below the level of detection or quantification.

Additional comments

3. There are multiple references throughout the report to “DIs” or "synthetic DIs” without specifying whether these are intended to refer to DI genomes or DI particles. Please specify in each case the intent.

Language

1) Lines 16-17 could be reworded for greater clarity - (in order to replicate and infect other cells, viruses must…)
2) Also in lines 16-17, clarify precisely what is being referred to as “they” (i.e. viruses, virions, etc.)
3) In lines 18-20, sentence could be split into two, creating a more succinct flow that makes it easier for the audience to follow
4) In line 23, specifically in the part that states “enable it to be replicated and packaged”- what is being referred to as “it”? Perhaps more specification would make this part clearer
5) Verbose sentence in lines 38-41; difficult to follow, may consider restating so that sentence ends after the phrase “full length viruses” and then proceed to elaborate on what would be needed for replication and packaging
6) The word “they” should be removed from line 42; end sentence after “packaging” in line 42 and begin next sentence with “given” for better flow and clarity
7) Lines 45-50 should be broken up into multiple sentences → transition into SARS-CoV-2 should be marked by a new sentence, as the information provided in this sentence is quite dense
8) Perhaps adding broader information regarding SARS-CoV-2 and its relation to the COVID pandemic would be beneficial, as it would provide more context
9) Remove comma in line 51, specify that “they” refers to the DI genome in line 51
10) After line 52, add a broader concluding sentence that specifies how replication and packaging of the DI genome could “impair the growth of the wild type (WT) virus”
11) In line 67, semicolon should be replaced with comma
12) Reword line 85 for greater clarity → (i.e. which were later infected with SARS-CoV-2)
13) Lines 190-194 are verbose; consider removing last part “as most of the RNA cannot replicate and is quickly degraded” because this is redundant (already stated in beginning of the sentence → “Because of the fast degradation of the synthetic RNA inside cells”)
14) In line 195, restate what “its” is referring to
15) In lines 195-196, “reduced the amount of SARS-CoV-2 by approximately half” could be stated in a way that better relays the decrease; moving the part that states “within 24 hours of transinfection” so that it is placed immediately after the colon may allow for greater clarity
16) In line 214, replace “must be” with “is likely” unless results prove this
17) Remove “it” from line 242, as it is not needed
18) In line 242, last part could be reworded for greater clarity, so that is effectively relays that the amount of packaging does not meet the threshold for detection
19) Sentence in lines 257-258, “By enabling replication…” can be omitted, as it is previously stated and thereby redundant
20) For better clarity, reword lines 261-263; brief discussion of why this “spill over” to humans could be significant might provide helpful context
21) Consider rewording lines 267-268 (i.e. DI therapy has been attempted to be used on viruses such as HIV and influenza; however, these viruses are not ideal…)
22) End sentence at the end of line 272
23) Figure 2 adds helpful context and allows for better understanding of what the primary goal of the study is
24) Methods are described thoroughly, yet concisely, and provides information that sufficiently relays primary aspects of the experimental design

---

## Round 0.2 · accepted · Accept

Thank you again for addressing the reviewer's concerns and questions and congratulations again.

Reviewer 1 ·

Basic reporting

NA

Experimental design

NA

Validity of the findings

NA

Additional comments

The author have not performed any new experiments, which would have helped address some of the questions of the reviewers. However, they have made significant changes to the manuscript to focus on 3 main points:
- SARS-CoV-2 DIs can replicate in coinfections (faster than the WT genome)
- SARS-CoV-2 DIs can be packaged in coinfection (as efficiently as the WT genome)
- SARS-CoV-2 DIs interfere with the WT genome (but still not known whether this would be enough for therapy)
The data in this manuscript convincingly address these 3 points. Furthermore, the authors have added many clarifications throughout the manuscript.